Syntactic- and morphology-based text augmentation framework for Arabic sentiment analysis

Duwairi Rehab 1 rehab@just.edu.jo
Abushaqra Ftoon 2
1 Department of Computer Information Systems, Jordan University of Science and Technology , Irbid , Jordan
2 Department of Computer Science, Jordan University of Science and Technology , Irbid , Jordan
Bolshoy Alexander
Electronic publication date: 2021 Apr 5
Publication date: 2021
Volume: 7
Electronic Location ID: e469
Received 2020 Nov 19; Accepted 2021 Mar 13
Copyright: © 2021 Duwairi and Abushaqra
Copyright year: 2021
Copyright holder: Duwairi and Abushaqra
License: This is an open access article distributed under the terms of the Creative Commons Attribution License, which permits unrestricted use, distribution, reproduction and adaptation in any medium and for any purpose provided that it is properly attributed. For attribution, the original author(s), title, publication source (PeerJ Computer Science) and either DOI or URL of the article must be cited.
License URL: https://creativecommons.org/licenses/by/4.0/

Keywords: Text augmentation, Sentiment analysis, Arabic text, Natural language processing, Morphology-based augmentation

Funding: Jordan University of Science and Technology, Jordan 20160150 This work was supported by Jordan University of Science and Technology, Jordan Grant No. (20160150). The funders had no role in study design, data collection and analysis, decision to publish, or preparation of the manuscript.

==============================
Arabic language is a challenging language for automatic processing. This is due to several intrinsic reasons such as Arabic multi-dialects, ambiguous syntax, syntactical flexibility and diacritics. Machine learning and deep learning frameworks require big datasets for training to ensure accurate predictions. This leads to another challenge faced by researches using Arabic text; as Arabic textual datasets of high quality are still scarce. In this paper, an intelligent framework for expanding or augmenting Arabic sentences is presented. The sentences were initially labelled by human annotators for sentiment analysis. The novel approach presented in this work relies on the rich morphology of Arabic, synonymy lists, syntactical or grammatical rules, and negation rules to generate new sentences from the seed sentences with their proper labels. Most augmentation techniques target image or video data. This study is the first work to target text augmentation for Arabic language. Using this framework, we were able to increase the size of the initial seed datasets by 10 folds. Experiments that assess the impact of this augmentation on sentiment analysis showed a 42% average increase in accuracy, due to the reliability and the high quality of the rules used to build this framework.

Introduction

Arabic language is considered the most widely spoken language among the Semitic languages (Weninger et al., 2011; Al-Huri, 2015). It is also one of the popular languages in the world. As the statistical studies in 2019 mentioned (Summary by Language Size, 2020), Arabic language is spoken by nearly 319 million people and is ranked the fifth between the world's languages after Chinese, Spanish, English and Urdu\Indian. Arabic native speakers are distributed throughout the Arab World as well as many other nearby areas. Also, Arabic has around 30 modern varieties or dialects; one of them is the standard form Modern Standard Arabic (MSA) (ISO 639, 2020). In 2012, the United Nations Economic and Social Commission for West Asia mentioned that Arabic language has achieved the highest growth rate on the Internet compared to other languages. Therefore, recently digital Arabic content on the internet became fairly large. However, this does not deny the reality that Arabic is considered a highly ambiguous language, especially when trying to analyze, classify and process Arabic data automatically.

Recently, many efforts have investigated the Arabic language whether to analyze the text (Mohammed, Crandall & Abdul-Mageed, 2012; Diab et al., 2007; Diab, Hacioglu & Jurafsky, 2004), parse statements (Green & Manning, 2010; Stanford Arabic Parser Tagset | Sketch Engine, 2018), analyze sentiment (Al-Humoud et al., 2015; Oussous et al., 2020; Ombabi, Ouarda & Alimi, 2020; Duwairi & Alfaqeh, 2016), recognize speech (Ahmed, Vogel & Renals, 2017; Alsharhan & Ramsay, 2019), translate statements (Galley et al., 2009; Al-Ibrahim & Duwairi, 2020), or detect depression (Bataineh, Duwairi & Abdullah, 2019); all these applications require the existence of comprehensive Arabic datasets. Building a dataset is not an easy task, as it requires tremendous effort, time and cost. Also, the recent application of machine learning and deep learning requires huge datasets which contain billions of records. For example, training a sentiment classifier using deep learning methods requires huge data properly labelled with polarity information. Therefore, an automatic expansion for Arabic datasets is very favorable, especially when knowing that manually collecting and annotating data are troublesome (Kobayashi, 2018).

Sentiment analysis is the task of processing data, mainly textual, in order to determine its polarity, i.e., positive, negative, or neutral (Oueslati et al., 2020; Al-Ayyoub et al., 2019). This task has several real-world applications with great impact on important domains such as business (Liu et al., 2007), politics (Ceron, Curini & Iacus, 2014; Ebrahimi, Yazdavar & Sheth, 2017), tourism (Gao, Hao & Fu, 2015) and marketing (Cambria et al., 2012). In general, sentiment analysis could be treated as an unsupervised learning task (Duwairi, Ahmed & Al-Rifai, 2015), supervised learning task (Duwairi & El-Orfali, 2014), or a hybrid of both. Unsupervised learning for sentiment analysis relies on sentiment lexicons. By comparison, supervised learning requires the existence of annotated or labelled data to train the classifiers. Ideally, data labeling for sentiment analysis mandates that each instance must be assigned a label from: positive, negative, or neutral. This task of labelling is usually human-based and thus it is costly. Corpora for sentiment analysis are usually gathered from social media; and due to the multilingual nature of such media, several researchers directed their efforts towards multilingual sentiment analysis (Lo et al., 2017; Vilares et al., 2018; Esuli, Moreo & Sebastiani, 2020).

Deep learning has received unprecedented attention in recent years and provided state-of-the-art results in many fields including sentiment analysis (Zhai & Zhang, 2016; Tang, Qin & Liu, 2015a, 2015b; Zhou et al., 2015; Le & Mikolov, 2014). However, deep neural networks need large amounts of data to train and tune their parameters. Data augmentation is a technique for expanding the datasets, and it was paired with deep learning applications. It has been used successfully with vision data and recently has received attention with textual data. Augmenting data that was initially labelled for sentiment analysis involves generating new sentences relying on the existing ones. The simplest form is to use the synonyms of the words to create new sentences with the same labels as the original ones. In this work, a framework for extending the size of datasets that were originally labeled for sentiment analysis is presented. Specifically, in focus on the Arabic syntax, grammar and morphology to create new sentences with the same labels or opposite labels as explained in “Description of Framework”.

The syntax of Arabic language is complex (Kevin, 2001)—as several matching cases are possible between words in the same sentence, while in addition, each word has several synonyms. Therefore, it is possible to generate tens of variants for an Arabic sentence while preserving its meaning. This task can be automated if the system is able to parse the statement and link it to lexical resources. Parsing is the process where each word in the text is labeled with its part of the speech tag (Verb, Object, Subject, etc.). However, parsing is not a simple process especially for Arabic language where the structure and order of the words are not specified. The Natural Language Processing Group at Stanford University has built an open-source parser (The Stanford Natural Language Processing Group, 2018). Stanford Parser provides a set of natural language processing functions. Mainly, it was built for English; later on, many developers have carried out extensive work to improve the code and the grammatical rules to make it more comprehensive. As a result, this parser has been extended to include languages other than English, such as Chinese, German, Italian and Arabic. The parsing tool takes a text file as input and generates the base forms of words, normalizes and interprets dates, times and numeric quantities. Finally, it analyzes the grammatical structure of the sentences. The output of the parsing process can be presented in several forms, such as phrase structures, trees, or dependencies.

For building the framework, initially the Stanford Arabic Parser was used to generate the parse trees of Arabic sentences. Afterwards, the augmentation rules generated were used on these trees, to generate several equivalent parse trees for the original sentences utilizing Arabic morphology, syntax, synonyms and negation particles. These augmentation rules can be broadly divided into: (1) rules which alter or swap branches of the parse trees as per Arabic syntax and thus generate new sentences with the same labels; (2) rules which generate new parse trees by utilizing the synonyms of words in these sentences, and also generate new sentences with the same original labels; (3) rules which insert negation particles into the sentences and thus generate new sentences with opposite labels. It is worth mentioning here that the work in this paper addresses text augmentation for sentiment analysis. This means that the labels of the investigated sentences are either neutral, positive or negative. Applying the sets of rules described in (1) or (2) above will generate new sentences with the same labels as the input sentences. By comparison, applying the set of rules described in (3) as aforementioned, generates new sentences with opposite labels to the input sentences. Experiments proved the viability and effectiveness of the augmentation framework by running three experiments using three datasets. The size of the original datasets substantially increased and the generated sentences were of high quality.

The rest of this article is organized as follows: “Related Work” briefly describes the related literature works. “Arabic Language Properties” explains the properties of Arabic language. “Description of Framework” explains the design of the transformation rules which are the core of the augmentation framework. “Negation” describes the implementation of the framework. “Evaluation” demonstrates the experiments which were carried out to assess the effectiveness of the proposed work. And finally “Conclusion” summarizes the conclusion of this work.

Related work

This section describes related studies which have utilized Arabic WordNet as a component of frameworks. It also describes the related work which addresses data augmentation.

Arabic WordNet

WordNet (Miller et al., 1990) is a large linguistic database, or hierarchical dictionary, which was initially developed for the English language. It has been very useful for the fields of computational linguistics and Natural Language Processing (Miller & Fellbaum, 2007). Because of its structure, the WordNet differs from other standard dictionaries, where it groups words based on their meanings. The English WordNet lexicon (Miller, 1995) is divided into syntactic categories that contain (nouns, verbs, adjectives and adverbs). It should be noted here that function words are deleted. However, WordNet grouped synonyms using the meaning (thesaurus) rather than the form (dictionaries). It also represents words redundantly—where a given word may appear in noun, verb and adverb syntactic categories. The WordNet consists of four parts (Miller et al., 1990): (1) lexicographers source files; (2) the tool to convert these files into the lexical database; (3) the lexical database; (4) software tools that are used to access the database.

WordNet has been very useful as it was used to build many Natural Language Processing applications, Information Retrieval, term expansion and document representations (Fellbaum & Vossen, 2007). For example, Varelas et al. (2005) compared the performance of using single ontology and different ontologies for the semantic similarity methods. Single ontology experiments were performed using the WordNet and it showed better performance in the results.

However, many efforts have been reported to adapt WordNet for other languages, such as WordNets for European languages (Vossen, 2004) and French and Slavonian WordNets (Sagot & Fišer, 2021). By comparison, Arabic WordNet (Elkateb et al., 2006) used the same development approach for word representation of Princeton WordNet to keep it compatible with other Word-Nets’ structures. Arabic WordNet is a lexical database for MSA, with two main linguistic categories (verbs and nouns). First, the important concepts that represent the core WordNet were extracted, then specific concepts for the Arabic language were developed along with other concepts that were manually translated to the most convenient synset from other languages. It was developed using MySQL and XML (Elkateb et al., 2006). Finally, the Arabic WordNet ended up with 11,270 synsets (2,538 verbs, 7,961 nominal, 110 adverbs and 661 adjectives) with 23,496 Arabic expressions. Table 1 presents detailed information about the statistical properties of Arabic WordNet.

Table 1 Statistical properties of Arabic WordNet.

	Unique strings	Synsets	
Noun	13,330	7,961	
Verb	5,595	2,536	
Named entities	1,426	1,155	
Broken plurals	405	126	
Total	20,756	11,778	

Several researchers have targeted extending Arabic WordNet. For example, in the work reported in Alkhalifa & Rodríguez (2009, 2010), the authors automatically extracted named entities from Arabic Wikipedia. Subsequently, they attached these entities as instances to the synsets of Arabic WordNet and finally created a link to their counterparts in English WordNet. Moreover, Badaro, Hajj & Habash (2020) introduced an automatic method for expanding Arabic WordNet—where they formulated the problem as a link prediction problem.

Shoaib et al. (2009) used the relationships in Arabic WordNet in order to build a model for semantic search in the Holy Quran. The proposed model improved searching and retrieving of the related verses from the Holy Quran without mentioning a specific keyword in the query. The model works in two stages. Namely, it identifies one sense of the query word using Word Sense Disambiguation, then it extracts out all the synonyms of the identified sense of the word. AlMaayah, Sawalha & Abushariah (2016) have also worked on the Holy Quran, where the researchers have built a model that extracts the synonyms and builds the Quranic Arabic WordNet. This net was built based on the Boundary Annotated Quran Corpus, lexicon resources, and traditional Arabic dictionaries. The final model was able to link the Holy Quran words that have the same meaning and generate sets of synsets using the vector space model. The Quranic Arabic WordNet has 6,918 synsets from 8,400 unique word senses. In other studies, the researches have tried to extract semantic relationships between words, and provide models to represent ontological relations for the Arabic content on the internet. These representations are useful to facilitate the analyses and processing of Arabic text. Al Zamil & Al-Radaideh (2014) have used the semantic features that were extracted from the text along with syntactic patterns of relationships to provide models that are able to automate the process of ontological relations extraction. The extracted features are used to construct generalized rules which were used to build a classifier. The classifier presents each concept with its designated relationship label.

Data augmentation

Data augmentation is a technique that is used to increase the size of datasets and preserve the labels at the same time. It became popular with deep learning networks as they require training on huge datasets to secure high accuracies (Krizhevsky, Sutskever & Hinton, 2012; Szegedy et al., 2015; Jaitly & Hinton, 2013; Ko et al., 2015). Extending the size (number of samples) in a dataset, especially for under-represented classes, is mainly depended on generating perturbed replicas of the class samples. This technique has proved its success in image classification such as the work reported in Krizhevsky, Sutskever & Hinton (2012), Tran et al. (2017) and Irsheidat & Duwairi (2020); 3D pose estimation as reported in Rogez & Schmid (2016); speaker language identification as described in Keren et al. (2016); recognition of audio-visual effect (Tzirakis et al., 2017); and the classification of the environmental sound (Salamon & Bello, 2017).

On the other hand, data augmentation is limited when dealing with textual data. This is due to the very difficult definition and standardization of specific rules or transformations that preserve the meaning of the produced textual data (Kobayashi, 2018). Basically, the main approach that works to increase the size of textual data, and preserves text meaning, is to use the synonyms of words, relying on lexical resources such as WordNet.

The works reported in Zhang, Zhao & LeCun (2015) and Wang & Yang (2015) have used a synonyms-based approach for augmenting textual data. As the synonyms are very limited, the proposed sentences are not very different and numerous from the original texts. Therefore, Kobayashi (2018) has proposed the contextual augmentation method, which is a state-of-the-art method to augment words, and produce more varied sentences. The author used words predicted by the bidirectional language model (LM) instead of using synonyms. The proposed approach was able to present a wide range of substitute words and it has been tested with two classifiers using recurrent or convolutional neural networks where it improves the overall performance. Rizos, Hemker & Schuller (2019) targeted extending a text used for hate speech detection relying on synonyms lists, wrapping the word token around the padded sequence, and finally applying class-based conditional recurrent neural language generation. The authors state that they achieved a 5.7% increase on Macro-F1 and a 30% in recall when extending the datasets using their three text extensions methods.

The work reported in Sharifirad, Jafarpour & Matwin (2018) has described a framework for augmenting tweets based on ConceptNet and Wikidata. The authors suggested two methods for improving the quality of tweets by first appending terms extracted from ConceptNet and Wikidata to the existing tweets but not increasing their numbers. Secondly, they generated new tweets by replacing words or terms in the original tweets with terms extracted from ConceptNet and Wikidata. This approach is close to the approaches which utilize synonyms.

In a similar study, Kolomiyets, Bethard & Moens (2011) replaced the headwords with a substitute word predicted from the Latent Words in the language model. The authors only used the top k score words as a substitute. Mueller & Thyagarajan (2016) substituted random words in sentences with their synonyms to generate new sentences. Subsequently, they trained a siamese recurrent network to compute the similarity between sentences. Wang & Yang (2015) employed word embedding to increase the size of the training data. Specifically, they replaced a given word with its nearest neighbor word vector.

As it can be seen from the above literature, most of the existing augmentation techniques address image or audio data and less work addresses text augmentation. In this regard, it should be mentioned that no work addresses Arabic text augmentation. The current proposed framework is substantially different from text augmentation which relies on the replacement of words by their synonyms. On the other hand, it utilizes the rich syntax and grammar of the Arabic language in order to generate transformation rules, that are subsequently used to generate new sentences based on seed sentences.

Arabic language properties

Arabic language is one of the Semitic languages. It consists of 28 basic letters. Several Arabic letters change their shapes based on their location in the word. For example, the letter (س) has the shape (ســ) when it is located at the beginning of the word, the shape (ـسـ) when it is located at the middle of the word, (ـس) when it is located at the end of the word but connected to the previous letter, and (س) when it comes at the end of the word but disconnected from the previous letter. Arabic is an inflectional language that is written from right to left. The following three subsections provide background about Arabic language.

Arabic morphology

Morphology is the structure of words. The morphology of Arabic language is complex but systematic—where there are two ways to build a word in Arabic: derivation and agglutination. The derivation is a way of generating stems from a list of roots; based on three basic letters (ف، ع، ل) for trilateral roots. For example, by using the root word “درس” that rhymes with “فعل” one can generate the following stems:Study “darasa دَرَسَ ”

Scholar “dAris دارِس ”

Lesson “daros دَرْس ”

Teacher “mudaris مُدَرِس ”

Schools “mdAris مدارِس ”

School “madrasop مَدرَسْة”

Study “mudArasop مُدارَسْة”

The second way to build words in Arabic language is agglutination. In this way, the words are built by adding affixes to the word. These affixes could be prefixes at the beginning of words such as (است، تم ، ت، ان), infixes in the middle of the word (such as ا), or suffixes at the end of the word such as (ة ، اء ، ان).

Arabic syntax

In Arabic scripts, the sentence has two types or categories (nominal and verbal). Each type has its own grammar and rules. The nominal sentence, in Arabic, consists of a subject (Almubtada) and predicate (Alkhabar). The normal order is that the subject is followed by the predicate but in certain cases, it is allowed to swap them (e.g. the sentence “أنْت َ مجتهدٌ ” which means “You are diligent” could be “مجتهدٌ أنْت َ‏”). The subject in the nominal sentence can be Noun, Pronoun or Number while the predicate can be Singular Noun, Adverb, Preposition, Nominal sentence, or Verbal Sentence.

The verbal sentence in Arabic, like in many other languages, consists of Verb (V), Subject (S) and Object (O) without a specific order, which means that the order of verbal sentences could be: VSO, VOS, SVO or VOS. Additionally, in Arabic language diacritics, prefixes and suffixes are used to represent gender. Therefore, the absence of diacritics can create ambiguity and might change the meaning.

Diacritics

One of the Arabic language features is the diacritics that are written above or underneath its letters. Diacritics are small vowel marks that represent three short vowels (a, i, u). They are used to regulate and control the letters and pronunciation. Therefore, diacritics have a huge effect on the text and its meaning, removing them may lead to morphological-lexical and morphological-syntactical ambiguities. For example, the word (nEm) (نعم) has the meaning ‘Yes’ if it was written (naEom نَعْم), while it means ‘graces’ if it was written (niEm نِعم). The basic diacritics of Arabic language are:Fatha: symbolized as an italic score on the top of the letter.

Dma: symbolized as a small (و) letter on the top of the letter.

Ksra: symbolized as an italic underscore on the bottom of the letter.

Sokon: symbolized as a small circle on the top of the letter.

Transformation rules definition

As a first step, clear definitions of Arabic grammar rules were specified. These rules include specifications for nominal sentences, verbal sentences, questions, verbs, adjectives, pronouns, prepositions, conjunctions and numbers. These defined grammar-based rules were represented using the Stanford Arabic parser tagset. Table 2 lists these tags in full details.

Table 2 Stanford Arabic parser tageset.

Tag	Description	Tag	Description	
ADJ	Adj	NNS	Noun, plural	
CC	Coordinating conjunction	NOUN	Noun	
CD	Cardinal number	PRP	Personal pronoun	
DT	Determiner	PRP$	Possessive pronoun	
DTJJ	Adjective with the determiner “Al” (ال)	PUNC	Punctuation	
DTJJR	Adjective, comparative with the determiner “Al” (ال)	RB	Adverb	
DTNN	Noun, singular or mass with the determiner “Al” (ال)	RP	Particle	
DTNNP	Proper noun, singular with the determiner “Al” (ال)	UH	Interjection	
DTNNPS	Proper noun, plural with the determiner “Al” (ال)	VB	Verb, base form	
DTNNS	Noun, plural with the determiner “Al” (ال)	VBD	Verb, past tense	
IN	Preposition or subordinating conjunction	VBG	Verb, gerund or present participle	
JJ	Adjective	VBN	Verb, past participle	
JJR	Adjective, comparative	VBP	Verb, non-3rd person singular present	
NN	Noun, singular or mass	VN	Verb, past participle	
NNP	Proper noun, singular	WP	Wh-pronoun	
NNPS	Proper noun, plural	WRB	Wh-adverb	

Table 3, on the other hand, summarizes the core concepts of this research—it depicts, in the second column, grammar rules for valid sentences in Arabic. The third column of Table 3 lists equivalent grammar rules which were derived from the original rules listed in the second column. The importance of these rules is that sentences that respect the grammar rules listed in the second column could be mapped to new sentences which fulfill the grammar rules listed in column 3, and still have the same label for the classifiers. The following statements show example sentences from Arabic which respect grammar rules in Table 3 and show how these sentences are transformed into new sentences, → means that the RHS of the rules are equivalent to the LHS:

Table 3 Transformation rules based on Arabic grammar.

ID	Original rules	Equivalent rules	
1	DTNN+ADJ	ADJ+DTNN	
2	NN+ADJ	ADJ+NN	
3	DTNN+NN	NN+DTNN	
4	NN+NN	NN+NN (swap)	
5	NN+DTNN	could not be changed	
6	DTNN+DTNN	DTNN+DTNN (swap)	
7	ADJ+ADJ	ADJ+ADJ (swap)	
8	PP+(NN+DTNN)	Place at the beginning and reverse the sentence.	
9	PP+(DTNN)	Place at the beginning and reverse the sentence.	
10	PP+(special character VB | NN)	Place at the beginning and reverse the sentence.	
11	Wh-prounoun+end of the sentences	Place at the beginning of sentences	
12	Special adverb+(NN | VB | (special character VB | NN))	(NN | VB | (special character VB | NN))+Special adverb	
13	Pronoun+(NN |VB | ADJ)	(NN |VB | ADJ)+Pronoun	
14	(NN|DTNN)+VB	VB+(NN|DTNN)	
15	NN+(Special-character+VB)	(special-character)+NN	
16	VB+(NN|DTNN)	(NN+DTNN)+VB	
17	VB+(Special-character+(NN|DTNN))	(Special-character+(NN|DTNN))+VB	
18	(Special-character+VB)+(Special-character+(NN|DTNN))	(Special-character+(NN|DTNN))+(Special-character+VB)	
19	(Special-character+VB)+(NN|DTNN)	(NN|DTNN)+(Special-character+VB)	
20	Special-character+(NN|DTNN))+VB	VB+(Special-character+(NN|DTNN))	
21	(Special-character+(NN|DTNN))+(Special-character+VB)	(Special-character+VB)+(Special-character+(NN|DTNN))	
22	CD+(NN|DTNN|VB)	could not be changed	
23	WH-Adverb+(NN|VB|DTNN|(Special-character+(NN|DTNN)) | (Special-character+VB))	(NN|VB|DTNN|(Special-character+(NN|DTNN)) | )Special-character+VB))+WH-Adverb	

RULE 1: DTNN+ADJ → ADJ+DTNN

Example: Alrjl mHbwb (الرجل محبوب) → mHbwb Alrjl (محبوب الرجل)

Parse: (ROOT (S (NP (DTNN الرجل)) (ADJP (JJ محبوب)))) → (ROOT (ADJP (JJ محبوب) (NP (DTNN الرجل))))

RULE 2: NN+ADJ → ADJ+NN

Example: mAlk rA}E (مالك رائع) → rA}E mAlk (رائع مالك)

Parse: (ROOT (S (NP (NNP مالك)) (ADJP (JJ رائع)))) → (ROOT (FRAG (NP (JJ رائع)) (NP (NNP مالك))))

RULE 3: DTNN+NN → NN+DTNN

Example: Alrjl $jAE (الرجل شجاع) → $jAE Alrjl (شجاع الرجل)

Parse: (ROOT (S (NP (DTNN الرجل)) (NP (NNP شجاع)))) → (ROOT (ADJP (JJ شجاع) (NP (DTNN الرجل))))

RULE 4: NN+NN → NN+NN(swap)

Example: AHmd Swth rA}E (احمد صوته رائع) → rA}E AHmd Swth (رائع احمد صوته)

Parse: (ROOT (S (NP (NNP احمد)) (NP (NN صوته) (JJ رائع)))) → (ROOT (FRAG (NP (JJ رائع)) (NP (NNP احمد) (NNP صوته))))

RULE 5: NN+DTNN → NN+DTNN

Example: Ebd AlrHmn xlwq (عبد الرحمن خلوق)→xlwq Ebd AlrHmn (خلوق عبد الرحمن)

Parse: (ROOT (FRAG (NP (NNP عبد)) (NP (DTNNP الرحمن)) (NP (NNP خلوق)))) → (ROOT (NP (NNP خلوق) (NNP عبد) (DTNNP الرحمن)))

RULE 6: DTNN+DTNN → DTNN+DTNN(swap)

Example: Ebd AlrHmn AlrHym (عبد الرحمن الرحيم) → AlrHym Ebd AlrHmn (الرحيم عبد الرحمن)

Parse: (ROOT (FRAG (NP (NNP عبد)) (NP (DTNNP الرحمن)) (NP (DTNNP الرحيم)))) → (ROOT (S (NP (DTNNP الرحيم)) (NP (NNP عبد) (DTNNP الرحمن))))

RULE 7: ADJ+ADJ → DTNN+DTNN (sawap)

Example: AlftAp Aljmylp mjthdp (الفتاة الجميلة مجتهدة) → AlftAp mjthdp Aljmylp (الفتاة مجتهدة الجميلة)

Parse: (ROOT (NP (DTNN الفتاة) (DTJJ الجميلة) (DTJJ مجتهدة))) → (ROOT (NP (DTNN الفتاة) (DTJJ مجتهدة) (DTJJ الجميلة)))

RULE 8: PP+(NN+DTNN) → place them in the beginning and reverse the sentence

Example: Ebr Alm$rf En $kr AlfSl (عبر المشرف عن شكر الفصل) → En $kr AlfSl Ebr Alm$rf (عن شكر الفصل عبر المشرف)

Parse: (ROOT (S (VP (VBD عبر) (NP (DTNN المشرف)) (PP (IN عن) (NP (NN شكر) (NP (DTNN الفصل))))))) → (ROOT (S (PP (IN عن) (NP (NN شكر) (NP (DTNN الفصل)))) (NP (NN عبر) (NP (DTNN المشرف)))))

RULE 9: PP+DTNN → place them in the beginning and revese the sentences

Example: bAsm yqdm $y}A mn AlfkAhAt (باسم يقدم شيئاً من الفكاهات) → mn AlfkAhAt bAsm yqdm $y}A (من الفكاهات باسم يقدم شيئاً)

Parse: ROOT (S (NP (NNP باسم)) (VP (VBP يقدم) (NP (NP (NN شيئا)) (PP (IN من) (NP (DTNNS الفكاهات))))))) → (ROOT (S (PP (IN من) (NP (NN الفكاهات) (NP (NNP باسم)))) (VP (VBP يقدم) (NP (NN شيئا)))))

RULE 10: PP+(Special character VB | NN)→ place them in the beginnig and reverse the sentence

Example: tSAdq mE Al*}Ab ElY >n ykwn f>sk mstEdA (تصادق مع الذئاب على أن يكون فأسك مستعد ) → Al*}Ab ElY >n ykwn tSAdq mE Al*}Ab f>sk mstEdA (على أن يكون تصادق مع الذئاب فأسك مستعداً)

Parse: (ROOT (S (VP (VBP تصادق) (NP (NN مع) (NP (DTNN الذئاب))) (PP (IN على) (NP (DTNN أن))) (S (VP (VBP يكون) (NP (NNP فأسك)) (ADJP (JJ مستعدا))))))) → (ROOT (S (PP (IN على) (NP (DTNN أن))) (VP (VBP يكون) (S (VP (VBP تصادق) (NP (NN مع) (NP (NNP الذئاب) (NNP فأسك))) (ADJP (JJ مستعدا)))))))

RULE 11: Wh-prounoun at the end of the sentences → Move it to the beginning

Example: njH Al*y *hb AlY Almdrsp (نجح الذي ذهب الى المدرسة) → Al*y *hb AlY Almdrsp njH (الذي ذهب الى المدرسة نجح)

Parse: (ROOT (S (VP (VBD نجح) (SBAR (WHNP (WP الذي)) (S (VP (VBD ذهب) (PP (IN الى) (NP (DTNN المدرسة))))))))) → (ROOT (S (SBAR (WHNP (WP الذي)) (S (VP (VBD ذهب) (PP (IN الى) (NP (DTNN المدرسة)))))) (VP (VBD نجح))))

RULE 12: Special adverb+(NN | VB | (special character VB | NN)) → (NN | VB | (special character VB | NN))+Special adverb

Example: AlEfw End Almqdrp (العفو عند المقدرة)→End Almqdrp AlEfw (عند المقدرة العفو)

Parse: (ROOT (NP (NP (DTNN العفو)) (NP (NN عند) (NP (DTNN المقدرة))))) → (ROOT (NP (NN عند) (NP (NP (DTNN المقدرة)) (NP (DTNN العفو)))))

Example: frH Alwld bxbr AlrHlp qbl >n y*hb (فرح الولد بخبر الرحلة قبل أن يذهب) → qbl >n y*hb frH Alwld bxbr AlrHlp (قبل أن يذهب فرح الولد بخبر الرحلة)

Parse: (ROOT (S (NP (NP (NN فرح) (NP (DTNN الولد))) (NP (NP (NN بخبر) (NP (DTNN الرحلة))) (NP (NN قبل) (NP (DTNN أن))))) (VP (VBP يذهب)))) → (ROOT (S (NP (NN قبل) (NP (DTNN أن))) (VP (VBP يذهب) (NP (NN فرح) (NP (DTNN الولد))) (NP (NN بخبر) (NP (DTNN الرحلة))))))

RULE 13: Pronoun+(NN |VB | ADJ) → (NN |VB | ADJ)+Pronoun

Example: hy tjyd AlxyATp (هي تجيد الخياطة) → tjyd hy AlxyATp (تجيد هي الخياطة)

Parse: (ROOT (S (NP (PRP هي)) (VP (VBP تجيد) (NP (DTNN الخياطة))))) → (ROOT (S (VP (VBP تجيد) (NP (PRP هي)) (NP (DTNN الخياطة)))))

Example: Ant rjl krym (أنت رجل كريم) → rjl krym Ant (رجل انت كريم)

Parse: (ROOT (S (NP (PRP انت)) (NP (NP (NN رجل)) (NP (NNP كريم))))) → (ROOT (NP (NP (NN رجل) (NP (PRP انت))) (NP (NNP كريم))))

Example: hy Al>jml (هي الأجمل) → hy Al>jml (الأجمل هي)

Parse: (ROOT (S (NP (PRP هي)) (NP (NNP الاجمل)))) → (ROOT (S (VP (VBP الاجمل) (NP (PRP هي)))))

RULE 14: (NN|DTNN)+VB → VB+(NN|DTNN)

Example: Alwld y>kl qlylA (الولد يأكل قليلاً) → y>kl Alwld qlylA (يأكل الولد قليلاً)

Parse: (ROOT (S (NP (DTNN الولد)) (VP (VBP يأكل) (NP (NN قليلا))))) → (ROOT (S (VP (VBP يأكل) (NP (DTNN الولد)) (NP (NN قليلا)))))

Example: bAsm yqdm $y}A mn AlfkAhAt (باسم يقدم شيئا من الفكاهات) → yqdm bAsm $y}A mn AlfkAhAt (يقدم باسم شيئا من الفكاهات)

Parse: (ROOT (S (NP (NNP باسم)) (VP (VBP يقدم) (NP (NP (NN شيئا)) (PP (IN من) (NP (DTNNS الفكاهات))))))) → (ROOT (S (VP (VBP يقدم) (NP (NNP باسم)) (NP (NP (NN شيئا)) (PP (IN من) (NP (DTNNS الفكاهات)))))))

RULE 15: NN+(Special-character+VB) → (special-character+vb)+NN

Example: Alwld <n ydrs ynjH (الولد إن يدرس ينجح) → <n ydrs Alwld ynjH (إن يدرس الولد ينجح)

Parse: (ROOT (S (NP (DTNN الولد) (DTJJ إن)) (VP (VBP يدرس) (S (VP (VBP ينجح)))))) → (ROOT (S (VP (VBD إن) (S (VP (VBP يدرس) (NP (DTNN الولد)) (S (VP (VBP ينجح))))))))

Example: bAsm lw ynAm mbkrA lA ytEb (باسم لو ينام مبكرا لا يتعب) → lw ynAm bAsm mbkrA lA ytEb 0(لوينام باسم مبكرا لا يتعب )

Parse: (ROOT (S (NP (NNP باسم)) (SBAR (IN لو) (S (VP (VBP ينام) (ADJP (JJ مبكرا))))) (PRT (RP لا)) (VP (VBP يتعب)))) → (ROOT (S (SBAR (IN لو) (S (VP (VBP ينام) (NP (NNP باسم)) (ADJP (JJ مبكرا))))) (VP (PRT (RP لا)) (VBP يتعب))))

RULE 16: VB+(NN|DTNN) → (NN+DTNN)+VB

Example: wqE Alwld ElY Al>rD (وقع الولد على الأرض) → Alwld wqE ElY Al>rD (الولد وقع على الأرض)

Parse: (ROOT (S (VP (VBD وقع) (NP (DTNN الولد)) (PP (IN على) (NP (DTNN الارض)))))) → (ROOT (S (NP (DTNN الولد)) (VP (VBD وقع) (PP (IN على) (NP (DTNN الأرض))))))

RULE 17: VB+(Special-character+(NN|DTNN)) → Special-character+(NN|DTNN))+VB

Example: Elmt >n AlwfA' Sfp EZymp (علمت أن الوفاء صفة عظيمة) → >n AlwfA' Elmt Sfp EZymp (أن الوفاء علمت صفة عظيمة)

Parse: (ROOT (S (VP (VBD علمت) (NP (NN أن) (NP (DTNN الوفاء))) (NP (NN صفة) (JJ عظيمة))))) → (ROOT (S (NP (NN أن) (NP (DTNN الوفاء))) (VP (VBD علمت) (NP (NN صفة) (JJ عظيمة)))))

RULE 18: (Special-character+VB)+(Special-character+(NN|DTNN)) → (Special-character+

(NN|DTNN))+(Special-character+VB)

Example: ln >Elm >n AlwfA' Sfp EZymp (لن أعلم أن الوفاء صفة عظيمة) →>n AlwfA' ln >Elm Sfp EZymp (أن الوفاء لن أعلم صفة عظيمة)

Parse: (ROOT (S (VP (PRT (RP لن)) (VBP أعلم) (NP (NN أن) (NP (DTNN الوفاء))) (NP (NN صفة) (JJ عظيمة))))) → (ROOT (S (NP (NN أن) (NP (DTNN الوفاء))) (VP (PRT (RP لن)) (VBP أعلم) (NP (NN صفة) (JJ عظيمة)))))

RULE 19: (Special-character+VB)+(NN|DTNN) → (NN|DTNN)+(Special-character+VB)

Example: ln >Elm Alwld n$yT (لن أعلم الولد نشيط) → Alwld ln >Elm n$yT (الولد لن أعلم نشيط)

Parse: (ROOT (S (VP (PRT (RP لن)) (VBP اعلم) (NP (DTNN الولد)) (ADJP (JJ نشيط))))) → (ROOT (S (NP (DTNN الولد)) (VP (PRT (RP لن)) (VBP اعلم) (ADJP (JJ نشيط)))))

RULE 20: (Special-character+(NN|DTNN))+VB → VB+(Special-character+(NN|DTNN))

Example: >n AlwfA' yEml mEk (ان الوفاء يعمل معك) → yEml >n AlwfA' mEk (يعمل أن الوفاء معك)

Parse: (ROOT (S (NP (NN أن) (NP (DTNN الوفاء))) (VP (VBP يعمل) (NP (NN معك)))) → (ROOT (S (VP (VBP يعمل) (NP (NN أن) (NP (DTNN الوفاء) (DTJJ معك))))))

RULE 21: (Special-character+(NN|DTNN))+(Special-character+VB) → (Special-character+VB)+

(Special-character+(NN|DTNN))

Example: >n AlwfA' lA yHlw mEk (ان الوفاء لا يحلو معك) → lA yHlw >n AlwfA' mEk (لا يحلو أن الوفاء معك)

Parse: (ROOT (S (NP (NN أن) (NP (DTNN الوفاء))) (VP (PRT (RP لا)) (VBP يحلو) (NP (NN معك))))) → (ROOT (S (VP (PRT (RP لا)) (VBP يحلو) (NP (NN أن) (NP (DTNN الوفاء) (DTJJ معك))))))

RULE 22: CD+(NN|DTNN|VB) → CD+(NN|DTNN|VB)

Example: njH TAlb fy Altwjyhy (نجح 15 طالب في التوجيهي) → fy Altwjyhy njH TAlb (في التوجيهي نجح 15 طالب)

Parse: (ROOT (S (VP (VBD نجح) (NP (CD 15) (NP (NN طالب))) (PP (IN في) (NP (ADJP (DTJJ التوجيهي))))))) → (ROOT (S (PP (IN في) (NP (ADJP (DTJJ التوجيهي)))) (VP (VBD نجح) (NP (CD 15) (NP (NN طالب))))))

RULE 23: WH-Adverb+(NN|VB|DTNN|(Special-character+(NN|DTNN)) | (Special-character+VB)) → (NN|VB|DTNN|(Special-character+(NN|DTNN)) | (Special-character+VB))+WH-Adverb

Example: kyf kl AlnAs y>klwn (كيف كل الناس يأكلون) → kl AlnAs kyf y>klwn (كل الناس كيف يأكلون)

Parse: (ROOT (SBARQ (WHADVP (WRB كيف)) (S (NP (NOUN_QUANT كل) (NP (DTNN الناس))) (S (VP (VBP يأكلون)))))) → (ROOT (S (NP (NOUN_QUANT كل)) (VP (NP (DTNN الناس)) (SBAR (WHADVP (WRB كيف)) (S (VP (VBP يأكلون))))))

Extensive experiments, showed that the Arabic Stanford parsing is not very accurate especially for the adverbs and negation words. This will adversely affect the system by generating wrong synonyms for the sentences. Therefore, there existed the need for declaring our own list of some adverbs, negation and special words because Stanford Parser does not assign the proper labels, as expected. These lists are presented in Table 4.

Table 4 List of special adverbs, negation and special words.

Word	Transliteration	Meaning	Word	Transliteration	Meaning	
قبل	qbl	Before	إن	Ena	that	
بعد	bEd	After	أن	Ana	that	
فوق	fwq	Above	فإن	fA’n	Then	
تحت	tHt	Under	لو	lw	If	
أسفل	Asfl	Down	كي	ky	So that	
أمام	A mAm	In front of	لكي	lky	in order to	
وراء	wrA'	behind	قد	qd	may	
أعلى	AElY	Top	لكن	lkn	But	
وسط	wsT	Center	لم	lm	did not	
عند	End	At	ما	mA	What	
خلف	xlf	behind	بعض	bED	Some	
شمال	$mAl	north	فقط	fqT	Just	
جنوب	jnwb	South	ليت	lyt	wish	
شرق	$rq	east	لعل	lEl	Might	
غرب	grb	West	ذي	*y	The	
يمين	ymyn	right	ليس	lys	Not	
يسار	ysAr	left	إذا	E *A	if	
كل	KL	Each				

Description of framework

The general framework is illustrated in Fig. 1. In the first step, a sentence with its label is passed to the system and it is converted to its canonical form (i.e. its parse tree). Secondly, multiple equivalent sentences (parse trees) are generated from the input sentence by replacing words with their synonyms. The synonyms are generated using Arabic WordNet. Thirdly, multiple variants of the sentences (parse trees), which were generated in step 2, are produced based on the transformation grammatical rules described in Table 3. The sentences generated in step 2 and step 3 all have the same label as the original input sentence. The Negation module is optionally called if we want to infuse negation particle into the generated sentences, and thus substantially increasing the number of generated sentences. The generated sentences from the Negation module have opposite labels to the input sentence and its variants. The following subsections describe, in detail, each developed module.

Figure 1 Framework for the proposed text augmentation tool.

Generate synonyms using Arabic WordNet

The Arabic WordNet browser is free and publicly available. It uses a locally-stored database of Arabic data in XML format—where words of the same meaning are linked through pre-defined lexical relations. Furthermore, the interface is modeled on the European language WordNet interface; hence, it contains the basic components with additional Arabic components. However, the performance of the Arabic WordNet is not satisfactory when compared with other WordNets. For example, the Arabic WordNet contains only 9.7% of the Arabic lexicon, while the English WordNet covers 67.5% of the English lexicon. Also, the Arabic WordNet synsets are linked only through hyponymy, synonymy and equivalence; correspondingly seven semantic relations are used in the English WordNet. However, since the main goal is generating the synonyms of the words, the limitation of the Arabic WordNet did not substantially affect the work. Also, to avoid the noise caused by diacritics, only the first top five synsets in each synonyms list were considered. Table 5 shows the first eight synsets for the Arabic word “Man – رجل”. As it can be seen from Table 5, the further we go deeper in generating synonyms, the higher the chance of generating wrong synonyms. The last two entries in Table 5 correspond to “leg and foot” and not “Man”.

Table 5 Examples of synsets for “Man – رجل” using Arabic WordNet.

Arabic	Transliteration	Meaning in English	
ذكر	*kr	Male	
عاشق	EA$q	Lover	
حبيب	Hbyb	Lover	
قرين	qryn	consort	
محبوب	mHbwb	Lover	
زوج	zwj	Husband	
قدم	qadam	Foot	
ساق	saAq	Leg	

Apply transformation rules to generate equivalent sentences

Employing the synonyms and the transformation rules, enables us to generate a huge number of sentences that are equivalent, in meaning and label, to the original input sentence. Every extracted synonym, using Arabic WordNet, creates a new sentence from the input sentence. Subsequently, these sentences are processed by the transformation module which selectively applies the proper transformation rules and generates even more sentences with the same meaning and label to the original sentence. Meaning, here, is defined in the loose sense of being suitable for sentiment analysis and is not from a linguistics perspective. From a linguistic perspective, synonymous sentences represent close meanings but not exactly the same. As an example, one can generate 47 sentences from the simple verbal sentence (“أكل الولد التفاحة”) (The boy ate the apple) using only the synonyms and transformation rules (i.e. without using the negation module which would generate even more sentences). Table 6 shows a sample of nine sentences generated from the example sentence (The boy ate the apple).

Table 6 Examples of some equivalent sentences generated from the statement (أكل الولد التفاحة).

Original: أكل الولد التفاحة	
WordNet Synonymous			Rules Possibilities	
Subject: الولد	1	أكل الولد التفاحة	
	2	الولد أكل التفاحة	
	3	أكل التفاحة الولد	
Subject: الشاب	4	أكل الشاب التفاحة	
	5	الشاب أكل التفاحة	
	6	أكل التفاحة الشاب	
Subject: الفتى	7	أكل الفتى التفاحة	
	8	الفتى أكل التفاحة	
	9	أكل التفاحة الفتى	
				

Generate parse trees

This module is responsible for generating parse trees for the original input sentence; and the generated sentences using Stanford Arabic parser tagset. With parse trees, it becomes easier to apply the suitable transformation rules to a given sentence, and it also facilitates the infusion of negation particles into sentences as described in the next section. Figure 2A depicts the parse tree of the sentence (أكل الولد التفاحة) (ate the boy the apple), while Fig. 2B shows the parse tree of the sentence (الولد أكل التفاحة) (the boy ate the apple). These two parse trees are equivalent.

Figure 2 Equivalent Parse Trees for the Same Sentence: (A) Verbal Form, (B) Nominal Form.

Negation

Negating a sentence in Arabic means inserting one of the negation particles used in Arabic into an affirmative sentence. Every negation particle, in Arabic, has its own rules in terms of the type of verbs or nouns it affects and in terms of the position in the sentence in which it is inserted. Negating a sentence will result in a new sentence that has an opposite meaning to the original input sentence. The label of the input sentence is also flipped from positive to negative. In addressing the negation problem, we adopted the Negation-aware Framework presented by Duwairi & Alshboul (2015), where the authors explore the effects of Arabic morphology on sentiment analysis. The study focused on five negations particles (لم، لن، لا، ما، ليس) that have been grouped into two categories based on their effect on the word as shown in Table 7.

Table 7 Negation particles and their effects in Arabic (Alkhalifa & Rodríguez, 2009).

Negation
Particle	BuckWalter
Transliteration	Category	Effect	
لم	lam	Group A	Affects the verb after the particle	
لن	Lan	Group A	Affects the verb after the particle	
لا	lA	Group A	Affects the verb after the particle	
ما	mA	Group A	Affects the verb after the particle	
ليس	laysa	Group B	Affects the following two nouns or affects the following verb.	
				

After defining the negation rules, the system is able to negate a set of sentences and generate all possible variations of these sentences as a result of inserting negation particles regardless if the sentences are nominal or verbal sentences. Table 8 shows an example of the output generated after applying negation to the positive verbal sentence (أعجب الولد الطعام) which means (The boy likes the food). As it can be seen from Table 8, this one sentence generates 10 sentences with opposite labels (i.e. the input sentence shows positive sentiment towards food while the 10 generated sentences convey negative sentiments towards food).

Table 8 Examples of negated Arabic verbal sentences.

Negation Particles	generated Sentences	
ما	ما أعجب الولد الطعام	
الطعام ما أعجب الولد	
لم	لم يعجب الولد الطعام	
الطعام لم يعجب الولد	
لن	لن يعجب الطعام الولد	
الطعام لن يعجب الولد	
لا	لا يعجب الطعام الولد	
الطعام لا يعجب الولد	
ليس	ليس يعجب الولد الطعام	
ليس الطعام يعجب الولد	

Evaluation

The following subsections describe, thoroughly, three experiments that were designed to test the accuracy of the proposed augmentation framework. Firstly, an assessment for the impact of the proposed framework on sentiment analysis was made. Secondly, we tested the correctness of each transformation rule. Finally, the accuracy of the Negation module was tested and formulated.

Experiment 1: classification of sentiment towards products

The aim of this experiment is to classify product reviews into positive, negative or neutral reviews. The focus of this experiment is not the classifier, but to assess the resulting accuracy of using the proposed framework when enlarging the size of the dataset. To perform the first experiment, we used a subset of a public dataset of product reviews (ElSahar & El-Beltagy, 2015) which contains 300 reviews written in Arabic collected from souq.com. The data was annotated with three labels (1: positive 0: Neutral, −1: negative). In this experiment, and before performing any changes on the original data, the data was tested using several supervised classifiers (Naive Bayes, K-nearest neighbor and Support vector machine). The data was divided into 70% for training and 30% for testing. All the classifiers used word embedding that is generated using AraVec with a dimension equals to 300 (Soliman, Eisa & El-Beltagy, 2017). After the training process for each classifier, the testing phase for each classifier’s performance and ability to classify the testing data was performed. Accuracy was used to assess the performance of each classifier. Accuracy is calculated by dividing the number of correctly classified reviews by the number of all reviews. The reported accuracy was equal to 54.18% using the SVM classifier, 49.99% using the Naïve Bayes classifier and 52.17% using the K-nearest neighbor classifier. Next, the data was fed into the augmentation tool where the size of the data was increased by almost 10 times. The generated dataset was tested using the same classifiers. In comparison with the previous results, the accuracy was increased by 42% on average. In details, the accuracy rates obtained by each classifier, using the augmented dataset, were 97% using the SVM, 87% using the NB and 91.66% using the K-nearest neighbor as illustrated in Fig. 3. This improvement was expected—as increasing the dataset size will subsequently improve the training process which leads to improving the overall performance of the classifier.

Figure 3 Accuracy rates using the original dataset and the augmented dataset.

Experiment 2: testing the efficiency of each transformation rule

The aim of this experiment was to test the accuracy of each transformation rule independently. To achieve this goal, it was preferable to design a small artificial dataset, which consists of 40 statements with positive sentiment, 32 statements with negative sentiment and 27 neutral statements. A total of 99 sentences were carefully designed to align with the 23 transformation rules. Each sentence was processed by the augmentation tool, and thus several sentences were generated for each input sentence. The generated sentences were manually inspected to test their validity. Rule accuracy is a measure that evaluates the ability of a given rule to generate correct and meaningful sentences. Rule accuracy is calculated by dividing the number of correct sentences generated by a given rule by the number of all sentences generated by that rule. “A correct sentence” means a grammatically correct and meaningful sentence. Table 9 shows the accuracy that was obtained for each rule. As can be seen from the table, all of the rules secured high accuracies. This means that the rules are capable of generating correct sentences. When examining the sources of error, we discovered that it was caused by improper synonymous words generated by the Arabic WordNet. It is important to note here that Arabic WordNet covers only 9.7% of the Arabic lexicon or vocabulary.

Table 9 Accuracy rate per transformation rule.

Rule 1	Rule 2	Rule 3	Rule 4	Rule 5	
77.5	89.3	90.07	94.28	97.53	
Rule 6	Rule 7	Rule 8	Rule 9	Rule 10	
91.50	96.36	94.4	100	100	
Rule 11	Rule 12	Rule 13	Rule 14	Rule 15	
98.79	96.96	100	100	100	
Rule 16	Rule 17	Rule 18	Rule 19	Rule 20	
100	100	97.5	100	100	
Rule 21	Rule 22				
100	100				

Experiment 3: the efficiency of negation rules

The goal of the third experiment is to assess the capability of the Negation module in order to generate correct sentences. A small artificial dataset which consists of 26 positive sentences and 24 negative sentences was created for this purpose. It should be mentioned here that the Negation module is responsible for inserting proper negation particles into the input sentences. Negation flips the polarity of the input sentence. This means that positive sentences will become negative and vice versa. All the resulted sentences from the Negation module are correct with their respective labels properly flipped.

Conclusion

In this study, a novel data augmentation framework for Arabic textual datasets for sentiment analysis was presented. In total, 23 transformation rules were designed to generate new sentences from the input ones. These rules were designed after carefully inspecting Arabic morphology and syntax. To increase the number of generated sentences for every rule, Arabic WordNet was used to swap the words with their respective synonyms. These rules preserve the labels of the input sentences. This means that if the input sentence has a positive label then the generated sentences also have positive labels. By the same token, if the label of the input sentence is negative, the labels of the generated sentences are also negative. The same is true for the neutral label. A Negation module was also designed to insert negation particles into Arabic sentences. This module inverts or flips the labels of the generated sentences, as this is the effect of negation particles on the polarity of statements. Experimentally, we tested the proposed framework by conducting three experiments. The first experiment has demonstrated the effect of increasing the dataset size, using the augmentation tool, on classification. As expected, the accuracy improved in all the classifiers. This indicates that the quality of the generated sentences was high. The second experiment was designed to test the accuracy of each transformation rule. An artificial dataset was designed for this purpose. All rules scored extremely high accuracies. The third and last experiment used an artificial dataset to assess the quality of the generated sentences from the Negation module. The experiment reveals that all generated sentences were correct with proper associated labels.

Supplemental Information

Supplemental Information 1 Subset of the Product Review Dataset.

Click here for additional data file.

Supplemental Information 2 Dataset used for Experiment 2 and Experiment 3.

Click here for additional data file.

Supplemental Information 3 Dataset used for Experiment 4.

Click here for additional data file.

Supplemental Information 4 Raw Data.

Click here for additional data file.

Additional Information and Declarations

Competing Interests

Author Contributions

Data Availability

The authors declare that they have no competing interests.

Rehab Duwairi conceived and designed the experiments, performed the experiments, analyzed the data, performed the computation work, prepared figures and/or tables, authored or reviewed drafts of the paper, and approved the final draft.

Ftoon Abushaqra performed the experiments, analyzed the data, performed the computation work, prepared figures and/or tables, and approved the final draft.

The following information was supplied regarding data availability:

The dataset is available in the Supplemental Files. The code includes all libraries to make it self contained and is available at FigShare: Duwairi, Rehab; Abushaqra, Ftoon (2021): CODE_Arabic_text_Augmentation.rar. figshare. Software. https://doi.org/10.6084/m9.figshare.14074268.v1

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
