# Peer review of "Syntactic- and morphology-based text augmentation framework for Arabic sentiment analysis"

_PeerJ Computer Science, doi:10.7717/peerj-cs.469_

## Round 0.1 · original submission · Major Revisions

The reviewers expressed very harsh criticisms nevertheless it seems very positive and constructive. I would recommend taking all reviewers' comments very seriously especially those related to recent publications/ Including those specifically mentioned by both reviewers. I would expect an extended discussion of all recent developments in Arabic sentiment analysis and deep learning.

Reviewer 1 ·

Basic reporting

Line 58: I do not agree with the word choice “infeasible”.
Annotation is feasible but it requires time and is expensive. It has already been done to other languages.

Line 109 WordNet is a hierarchical dictionary.
Some typing and grammatical errors:
- line 39 word should world.
- Line 85 describe should be described
- Line 109 the word resembles should be replaced by grouped
- Line 203 replace ‘, briefly, lists’ with summarize.
- Line 214 215 review the verb tenses
- Line 485 ‘is’ should bet ‘it’
- Line 506 ‘like’ should be ‘likes’
- Line 533, usage of ‘whereas’ is not correct.
- Line 574, “thee” should be “three”
Line 206: I am not sure this is correct and what it even means. You need to clearly clarify this point.
Line 227 to 233: the Latin form of the Arabic words is incorrect, and it is not consistent. The ‘a’ in Modarsh should be “A”. Also, the sh is not a good way to represent the “t marboutap”. I suggest you use the same form as seen in the rules (line 279 and on).

The quality of the figures seems to be low.
Is the reader expected to fill the gaps in Table 6?

Since your objective is data augmentation, there were several efforts for augmenting the Arabic WordNet and some of them were used to develop Arabic sentiment and emotion lexicons. These should be included in your related work:
- Musa Alkhalifa and Horacio Rodríguez. 2009. Automatically extending NE coverage of Arabic WordNet using Wikipedia. In Proceedings of the 3rd International Conference on Arabic Language Processing (CITALA’09)
- Musa Alkhalifa and Horacio Rodríguez. 2010. Automatically extending named entities coverage of Arabic WordNet using Wikipedia. Int. J. Inf. Commun. Technol. 3, 3 (2010), 20—36
- Benoît Sagot and Darja Fišer. 2011. Extending WordNets by learning from multiple resources. In Proceedings of the 5th Language and Technology Conference (LTC’11).
- Badaro, Gilbert, Hazem Hajj, and Nizar Habash. "A Link Prediction Approach for Accurately Mapping a Large-scale Arabic Lexical Resource to English WordNet." ACM Transactions on Asian and Low-Resource Language Information Processing (TALLIP) 19, no. 6 (2020): 1-38.

Experimental design

Rule 4 does not seem to work. The adjective is now describing ‘AHmd’ instead of his voice.
Same for Rule 5, rule 7, rule 18, 19, 21, The class label is remaining the same because the content did not change. Therefore, any manipulation of the constituents would result in the same sentiment class label.

How did you deal with the discrepancy in the transliteration between the Stanford tool and Arabic WordNet?

All experiments setups suffer from very ambiguous descriptions. For example, experiment 1, what were the features used as input to the different classifiers? What were the percentages of training/testing data?

I am not sure how the negation rules resulted in fully correct sentences especially that in the simple example provided for the boy likes the food, there are incorrect sentences with the usage of ‘laysa’. Moreover, how as the verb conjugation handled?

Validity of the findings

The analysis of the results is very shallow. While the first part of the paper has good content about the Arabic complexity, the second part suffers from lack of details.

Additional comments

Unfortunately, the current shape of the paper suffers from multiple deficiencies as mentioned in the sections above.
In summary, justifications for the selected rules need to be justified. Would a simple shuffling of the words result in the same accuracy improvement in experiment 1?
All experiments require further details and clarity. Testing the efficiency of the approach should be performed on a larger dataset. More detailed analysis of the results is required.

Reviewer 2 ·

Basic reporting

see my comments below

Experimental design

see my comments below

Validity of the findings

see my comments below

Additional comments

The manuscript is centered on an interesting topic. Organization of the paper is good and the proposed method is quite novel.

The manuscript, however, does not link well with recent literature on sentiment analysis appeared in relevant top-tier journals, e.g., the IEEE Intelligent Systems department on "Affective Computing and Sentiment Analysis". Also, latest trends in multilingual sentiment analysis are missing, e.g., see Lo et al.’s recent survey on multilingual sentiment analysis (from formal to informal and scarce resource languages) and Oueslati et al.’s review of sentiment analysis research in Arabic language. Finally, check recent resources for multilingual sentiment analysis, e.g., BabelSenticNet.

The manuscript presents some bad English constructions, grammar mistakes, and misuse of articles: a professional language editing service is strongly recommended (e.g., the ones offered by IEEE, Elsevier, and Springer) to sufficiently improve the paper's presentation quality for meeting PeerJ CS’ high standards.

Finally, double-check both definition and usage of acronyms: every acronym should be defined only once (at the first occurrence) and always used afterwards (except for abstract and section titles). Also, it is not recommendable to generate acronyms for multiword expressions that are shorter than 3 words (unless they are universally recognized, e.g., AI).

---

## Round 0.2 · accepted · Accept

I would highly recommend consideration of the comments of Reviewer 2.

Reviewer 1 ·

Basic reporting

Comments were addressed.

Experimental design

Comments were addressed.

Validity of the findings

Comments were addressed.

Additional comments

The previous comments were addressed and the quality of the article has been significantly improved.

Reviewer 2 ·

Basic reporting

see below

Experimental design

see below

Validity of the findings

see below

Additional comments

The authors have addressed most of the concerns raised by the reviewers and their revisions have substantially improved the manuscript. However, there are still some minor issues to be addressed, namely:
1) presentation is better but there is still room for improvement
2) some references have incomplete compilation, e.g., missing page numbering